# Becoming the temporary surgeon: A grounded theory examination of anaesthetists performing emergency front of neck access in inter-disciplinary simulation-based training

**Sergio A. Silverio**[1,2,3]*, **Hilary Wallace**[4], **William Gauntlett**[5], **Richard Berwick**[4,6], **Simon Mercer**[4,7], **Ben Morton**[8,9,10], **Simon N. Rogers**[11,12], **John E. Sandars**[12], **Peter Groom**[4], **Jeremy M. Brown**[12]

1 Department of Women & Children's Health, School of Life Course Sciences, King's College London, London, United Kingdom, 2 Elizabeth Garrett Anderson Institute for Women's Health, Faculty of Population Health Sciences, University College London, London, United Kingdom, 3 Department of Psychology, Institute of Population Health, University of Liverpool, Liverpool, United Kingdom, 4 Anaesthesia and Theatres Department, Aintree University Hospital, Liverpool University Hospitals NHS Foundation Trust, Liverpool, United Kingdom, 5 The Jackson Rees Department of Anaesthesia, Alder Hey Children's NHS Foundation Trust, Liverpool, United Kingdom, 6 Pain Research Institute, Institute of Translational Medicine, University of Liverpool, Liverpool, United Kingdom, 7 Medical Education Department, Aintree University Hospital, Liverpool University Hospitals NHS Foundation Trust, Liverpool, United Kingdom, 8 Department of Clinical Sciences, Liverpool School of Tropical Medicine, Liverpool, United Kingdom, 9 Critical Care Department, Aintree University Hospital, Liverpool University Hospitals NHS Foundation Trust, Liverpool, United Kingdom, 10 Malawi-Liverpool-Wellcome Trust Clinical Research Programme, Queen Elizabeth Central Hospital, Blantyre, Malawi, 11 Oral and Maxillofacial Surgery Department, Aintree University Hospital, Liverpool University Hospitals NHS Foundation Trust, Liverpool, United Kingdom, 12 Health Research Institute, Faculty of Health & Social Care, Edge Hill University, Ormskirk, United Kingdom

* Sergio.Silverio@kcl.ac.uk

## Abstract

The time-critical 'can't intubate, can't oxygenate' [CICO] emergency post-induction of anaesthesia is rare, but one which, should it occur, requires Anaesthetists to perform rapid emergency front of neck access [FONA] to the trachea, restoring oxygenation, and preventing death or brain hypoxia. The UK Difficult Airway Society [DAS] has directed all Anaesthetists to be trained with surgical cricothyroidotomy [SCT] as the primary emergency FONA method, sometimes referred to as 'Cric' as a shorthand. We present a longitudinal analysis using a classical approach to Grounded Theory methodology of ten Specialist Trainee Anaesthetists' data during a 6-month training programme delivered jointly by Anaesthetists and Surgeons. We identified with a critical realist ontology and an objectivist epistemology meaning data interpretation was driven by participants' narratives and accepted as true accounts of their experience. Our theory comprises three themes: 'Identity as an Anaesthetist'; 'The Role of a Temporary Surgeon'; and 'Training to Reconcile Identities', whereby training facilitated the psychological transition from a 'bloodless Doctor' (Anaesthetist) to becoming a 'temporary Surgeon'. The training programme enabled Specialist Trainees to move between the role of control and responsibility (Identity as an Anaesthetist), through

**Data Availability Statement:** Data are subject to ethical restrictions imposed by the University Research Ethics Committee and Hospital Research and Development Department, because they contain potentially identifying or sensitive information. Data, therefore, cannot be shared publicly, however, are available upon reasonable request from the Health Research Institute at Edge Hill University (contact via: fohscresearch@edgehill.ac.uk), for researchers who meet the criteria for access to confidential data.

**Funding:** PG: This work was supported by a small project grant from the Difficult Airway Society awarded via the National Institute for Academic Anaesthesia. It was also supported by a grant from the Mersey School of Anaesthesia Charity.

**Competing interests:** Sergio A. Silverio (King's College London) is supported by the National Institute for Health Research Applied Research Collaboration South London [NIHR ARC South London] at King's College Hospital NHS Foundation Trust. The views expressed are those of the authors and not necessarily those of the NIHR or the Department of Health and Social Care. The authors have declared no potential conflicts of interest with respect to the research, authorship, and/or publication of this article. This does not alter our adherence to PLOS ONE policies on sharing data and materials.

**Abbreviations:** CICO, Can't Intubate, Can't Oxygenate'; DAS, Difficult Airway Society; FONA, Front of Neck Access; NAP4, 4[th] National Audit Project; RCoA, Royal College of Anaesthetists; SCT, Surgical Cricothyroidotomy; UK, United Kingdom.

self-described 'failure' and into a role of uncertainty about one's own confidence and competence (The Role of a Temporary Surgeon), and then return to the Anaesthetist's role once the airway had been established. Understanding the complexity of an intervention and providing a better insight into the training needs of Anaesthetic trainees, via a Grounded Theory approach, allows us to evaluate training programmes against the recognised technical and non-technical needs of those being trained.

# Introduction

## Background

A 'can't intubate, can't oxygenate' [CICO] scenario is a rare, but life-threatening event which occurs when an Anaesthetist is unable to pass a breathing tube into a patient and also cannot deliver oxygen to the patient via other means. Management of a CICO scenario requires rapid surgical access to the patient's trachea via the front of the neck in order to re-establish oxygenation. Faced with this emergency situation, the rates of success by Anaesthetists are low and outcomes are often tragic [1, 2]. The Royal College of Anaesthetists [RCoA] 4th National Audit Project [NAP4] highlighted deficiencies in emergency airway management [3], promoting a shift in the approach to managing a CICO scenario. The Difficult Airway Society [DAS] guidelines [1] now advocate a surgical cricothyroidotomy [SCT] for emergency front of neck access [FONA; also known as 'Plan D'], instead of the variety of kits once available for percutaneous needle access. The SCT method, requires an Anaesthetist to perform either a horizontal stab incision for a palpable cricothyroid membrane or an 8-10cm longitudinal incision for a non-palpable membrane. This has simplified what had historically been a complex choice of techniques available to the Anaesthetist and it is felt to have reduced operator indecision [1]. For any Anaesthetist confronting a CICO scenario it is a frightening and uncertain event [2].

Currently, the management of CICO is seen as a core competency for all Anaesthetic Trainees [4]. It requires attempting a surgical technique which is very different to their normal, technical skillset and in the context of a patient facing a life-threatening airway emergency. Though guidelines and simplifications are felt to have helped, the challenge for Anaesthetists is how to establish, maintain and improve the skills required to manage a CICO scenario when the scenario itself is extremely rare and, consequently, first-hand experience is limited [2]. Education and rehearsal are highlighted by both the RCoA and DAS as key to better management of CICO. Simulation, in particular, has been identified to increase skill retention for emergency airway training [5], whilst routine training in the clinical environment promotes retention [6]. Simulation has also been recognised as especially useful in preparing for the use of SCT in emergency FONA situations [6–8]. The acquisition of technical skill is only one part of successful preparation for a CICO scenario. Work from Timmermann and colleagues [9 p5] identified reticence in performing FONA is common, and that procedure success is *"dependent upon a clinician's willingness to implement it"*. Appropriate training is seen as crucial in developing this willingness; however, as yet there is no national, standard, training by which all Anaesthetists will be trained.

## Development of the simulation-based training

Within the UK NHS system, healthcare provision is organised into 'Trusts' which serve either a geographic area or provide a specialist function (such as mental health, or ambulance

services). A Trust can be made up of a single hospital or multiple hospitals. Existing literature highlights some of the technical and non-technical obstacles to the management of a CICO situation and the performance of FONA [10]. To date, however, there is no research which combines a thorough examination of these challenges, with an assessment of the effectiveness of a training intervention. We developed a new and innovative training programme for Specialist Trainee Anaesthetists ('Junior Doctors') on their clinical rotation to a local hospital, which we believe addresses many of the perceived challenges to FONA education (see Fig 1 for a flow diagram of the whole training programme).

Training was inter-disciplinary (bringing together senior surgical and anaesthetic expertise) and comprised: One-to-one tutoring on the 2015 DAS guidelines and cognitive aids; repetition of SCT instruction on an inanimate manikin; and surgical experience of performing elective surgical tracheostomies. It also included rehearsal of the manikin-based CICO scenario devised by our clinical team [PG, SNR, SM, BM, WG, HW, RB] which was an *in-situ*, real-time, reactive (including standardized: tracheal obstruction, falling SATS, and simulated bleeding when incision was made), role play exercise, undertaken at three timepoints during the training (baseline, 2-weeks, and 6-months). Training was completed within two weeks, however for the purposes of this medical education evaluation, we included a 6-month follow-up to assess retention of knowledge and skills. In combining these anaesthetic and surgical skills training, we believe we are providing a training regime which improves the preparedness of Anaesthetists to cope with, and manage a CICO scenario effectively.

## Evaluation of the training programme

The training was evaluated through the use of in-depth, semi-structured interviews [11], enabling a 'deep dive' into the psychological barriers and the change in patterns of identity, as well as the professional negotiations the clinicians on the CICO training programme experienced. By adopting a Grounded Theory approach [12], we uncovered a clearer picture of the challenges standing between the Anaesthetist and success in these emergency scenarios. Through this improved understanding we are better positioned to evaluate our current training provision, improve upon it, understand the psychological complexity of undertaking this training, and gain insight into the specific needs of Anaesthetists faced with this scenario.

## Grounded theory in clinical research

Grounded Theory is a commonly used method of analysis for qualitative data [13]. Increasingly, both qualitative data collection and Grounded Theory analysis is being used in healthcare research and health services evaluations [12, 14].

To name a few recent examples, specifically in terms of clinical identities, Grounded Theory has been demonstrated to produce high quality findings on how nursing staff reconcile their identities as specialist practitioners, teachers, and Nurses [15, 16], and in global studies, how clinicians learn [17], navigate their working environments [18–20], and negotiate their career-related errors and regrets [21–24]. It is with this reasoning, the complexity of the training intervention we were questioning, and the multi-disciplinary nature of our team (Psychologists, clinical anaesthetic and surgical Doctors, and Medical Educators), a Grounded Theory approach appropriate to cross-disciplinary health research was selected to ensure the most nuanced reflections were sought from participants.

## Aim

The study had two aims: 1) To explore the concerns, thoughts, and attitudes of trainee anaesthetists about the management of CICO and the performance of FONA over the 6-months

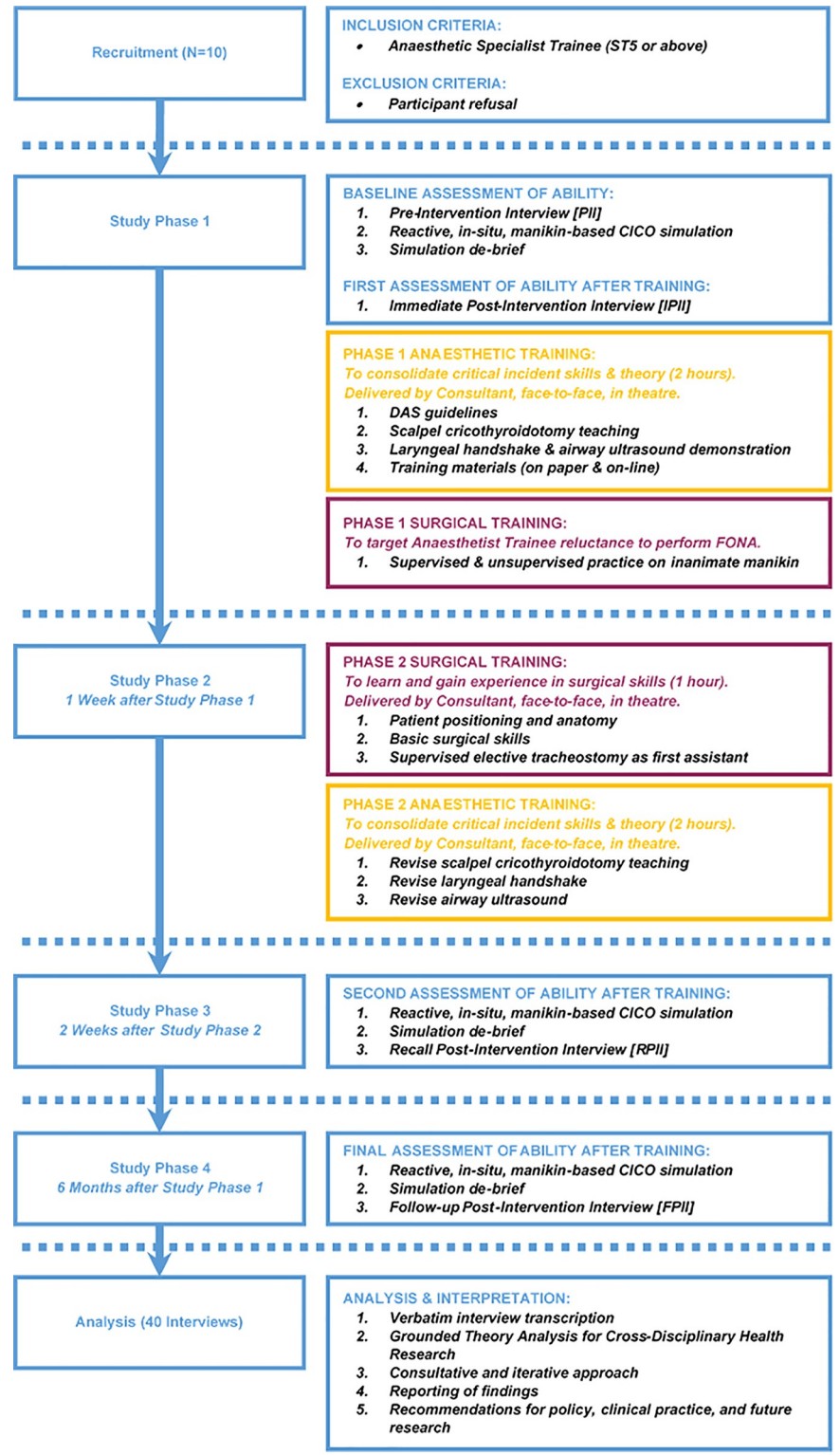

**Fig 1. Study phases.**

they participated in a training pilot; 2) To develop a theoretical model to explain how Anaesthetic Trainees view and engage with CICO scenarios and the impact, if any, of the training pilot.

## Research ethics

Ethical approvals were sought from and granted by Health Research Authority with IRAS project ID: 226720 and REC reference: 18/HRA/0122, with both Edge Hill University Faculty of Health and Social Care Research Ethics Committee and Aintree University Hospital Research and Development permissions, subsequently granted. All participants provided fully informed consent, by signing two copies of a written consent form, which were both witnessed by the researcher. Participants kept one copy and the research retained the other to keep in the study file.

## Methods

### Design

This study utilised a hybrid Grounded Theory analysis appropriate for cross-disciplinary health research of in-depth, qualitative interviews [12]. This hybrid approach to Grounded Theory methodology is based heavily on a classical Grounded Theory approach [25–27], but has elements of Straussian Grounded Theory [28]. This design was chosen to allow for a thorough consideration of a unique phenomenon–emergency FONA performed by SCT–in the wider professional context. In keeping with a Glaserian approach to Grounded Theory, we ensured a data-driven method of analysis, meaning we allowed the data to dictate the narrative, rather than imposing one on the data. We decided upon a participant-by-participant method of longitudinal analysis to develop a final theoretical model (see Silverio and colleagues [12] for a detailed explanation of the methodological process).

### Setting

The training programme was initiated by the Anaesthesia and Theatres Department and the Oral and Maxillofacial Surgery Department at a local University Hospital, specialising in head and neck surgery, based in the North West of England, United Kingdom. The *in-situ* simulations were video-recorded to use in feedback and for research purposes (as in Bleakley and colleagues [29]). The interviews were undertaken by local university academic researchers [SAS, JMB], and analysis was undertaken by both clinical and academic members of the research team.

### Participants

A total of 10 Specialist Trainee Anaesthetists [ST Anaesthesia] were recruited to the study, ranging in grade from ST5-ST7 (see Table 1). In the UK, 'Specialist Trainees' are fully qualified medical graduates also referred to as 'Junior Doctors', but who are in training for a specialised area of medicine under the supervision of a more Senior Doctor, which can take up to 8-years dependent on the speciality [30]. Sampling consisted of specialist trainees who were on their clinical rotation at the University Hospital, and participants were interviewed at four timepoints longitudinally (see Fig 1). All participants participated in all four interviews, and were interviewed at the same intervals over the course of the study: Pre-Intervention Interview [PII]; Immediate Post-Intervention Interview [IPII]; Recall Post-Intervention Interview [RPII; 2-week delay]; and Follow-Up Post-Intervention Interview [FPII; 6-month delay].

**Table 1. Participant information.**

| Identifier | Training Grade | Gender | Age (Years) |
|---|---|---|---|
| LH32 | ST5 | Female | 32 |
| LS34 | ST5 | Female | 34 |
| MC34 | ST5 | Female | 34 |
| PE30 | ST5 | Female | 30 |
| TW34 | ST5 | Female | 34 |
| CL31 | ST6 | Female | 31 |
| JH33 | ST6 | Male | 33 |
| SA32 | ST6 | Female | 32 |
| SF36 | ST6 | Female | 36 |
| AS35 | ST7 | Male | 35 |

## Data collection

Data were collected between April and October 2017 in a private office located on the site of the participating hospital. Interviews were conducted by an academic member of the research team, who was unknown to the participants and who was not medically trained to facilitate a naïve-researcher/expert-participant interaction [31]. Rigour was established in data collection, assessment, and analysis by following processes set-out by Mays and Pope [32]. These processes include reflecting on the research question for worth or relevance and clarity; ensuring an appropriate research design was selected for the question and providing a systematic 'audit trail' for how data were collected. Participants were asked a series of questions in each interview relating to their profession as an Anaesthetist; the CICO scenario and the emergency FONA by SCT procedure; their reflections of the training programme; and the perceived effect it had on the competence and confidence of their clinical practice. Data from the first interviews demonstrated aspects pertaining to psychological shifts in identity when conceptualising the CICO scenario, therefore subsequent interviews more thoroughly explored issues of identity. Interviews lasted between 20- and 60-minutes. The semi-structured interview schedules were adapted given the changing foci of participants' discussions at each previous timepoint, allowing for a data-driven exploration of participant's perspectives [12, 25, 33]. An example of an adaptation to the interview schedules was the increasing focus on identity by participants, leading to later timepoints (3 & 4) relating issues of practical confidence and competence performing simulated CICO scenario being questioned in relation to professional identity and personal development, rather than simply mechanical competence and self-professed confidence. Each of the forty interviews were transcribed verbatim and were then checked for meaning and accuracy by the interviewer, which also served as a re-familiarisation process. As per both Glaser & Strauss [26] and Silverio and colleagues [12], memo writings were recorded throughout the interview, re-familiarisation, and analytical processes, and were integral to code generation; theme development; and final theorisations about the data, the population interviewed, and the phenomenon explored. All data were managed manually.

## Data analysis

Analysis was led by one member of the team–a Psychologist [SAS]. All transcripts were also analysed by one of the clinical team [50% to each: HW, WG], and half of those analysed by both HW and WG were further analysed by a clinical education specialist [JMB], meaning 50% of data were analysed by psychologically-, clinically-, and education-focused members of the research team. This allowed for the analysis to satisfy psychological, education and training-, and clinical medicine perspectives similar to previous qualitative healthcare research

[34, 35]. However, this also allowed us to ensure any disciplinary biases were not being imposed on the data as we checked on each other's analysis and interpretation. This ensured participant data informed the final theory, and not the preconceived notions of the analysts [12]. Participants' interviews were analysed longitudinally, meaning each of the four interviews per participant were analysed in their entirety before moving onto analysis on the next participant [12]. This enabled analysis to depict changes in each participant over time as well as comparisons between each participants' complete dataset. In this respect, data can be collected longitudinally, but the dataset is still treated as a whole, and therefore not stratified by time-point, allowing for meaningful iterative analyses, in-line with classical Grounded Theory approaches. Analysis was iterative, meaning when comparing between transcripts, early transcripts were re-coded to take into account that early coding may not be as nuanced as coding which occurs after a number of transcripts have been analysed [27]. Analysis was also consultative, whereby analysts regularly met to discuss the data and debate the generation of themes, whilst resolving any differences of opinion about the resulting theory [33]. By analysing these data iteratively and in a consultative manner we were able to demonstrate our analytical approach as rigorous and ensure a high level of thematic concordance between different analysts' coding. This meant core concepts of the theory were present in each of our analyses, even if worded in a slightly different way [12]. It has been suggested that theme saturation can be acquired with relatively few participants if the data is useable and of high quality [36]. Our regular analytical discussions which referred to these models enabled us to deem our data of excellent quality. The team agreed meta-themes [36] were reached after 6-participants (24-interviews), with full saturation achieved by 9-participants (36-interviews). We were therefore satisfied we had recruited enough participants to be confident in our reported findings. No pre-conceived notion of the outcomes was made [26], but subsequent iterative thematic concepts could be mapped onto existing theoretical literature on emergency procedures, anaesthetics, and simulation training [28]. Coding was initially broad and open, with codes being assigned to transcripts using participants' own words. These codes were then compared, some were grouped and merged into more conceptual focused codes. Through yet more iterations of coding and with consultation amongst the analysis team [SAS, HW, WG, JMB] lower-order codes were either merged or split to form seven different super-categories: 'Anaesthesia as a Thoroughly Trained Specialism'; 'Situational Control as an Anaesthetist'; 'Being More Competent'; 'Feeling More Confident'; 'Procedure Seen as a Failure'; 'Causing Harm to Patients'; and 'Acceptance of Procedure as "Necessary"' (*see* Fig 2 for thematic diagram). Super-categories were subsequently collapsed into three themes, with the relational 'Anaesthesia as a Thoroughly Trained Specialism' and 'Situational Control as an Anaesthetist'; becoming *Theme 1*: *'Identity as an Anaesthetist'*. Super-categories: 'Procedure Seen as a Failure'; and 'Causing Harm to Patients' became *Theme 2*: *'The Role of a Temporary Surgeon'*; whilst the cyclical super-categories 'Being More Competent' and 'Feeling More Confident' were collapsed together, along with 'Acceptance of Procedure as "Necessary"' to form *Theme 3*: *'Training to Reconcile Identities'* (*see* Fig 3 for thematic diagram). Presented below are the most illustrative quotations representing each theme, with supplementary questions presented in Table 2. Quotations have been organised by timepoint, in-line with the longitudinal nature of our study design.

## Results

### Theme 1: 'Identity as an Anaesthetist'

Participants spoke across their interviews about their 'Identity as an Anaesthetist', especially in relation to how this significantly differed from that of the surgical role they were expected to

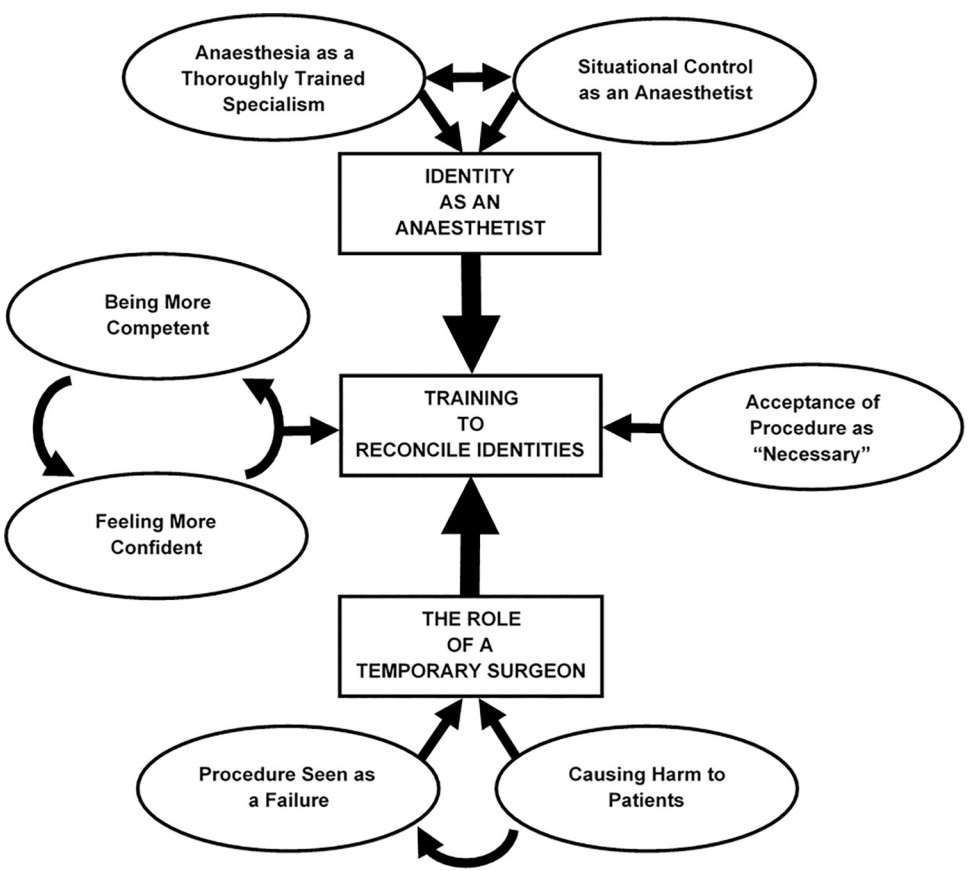

**Fig 2. Thematic diagram of super-categories.**

assume, but also in how it was characterised within the profession. Largely, participants saw their Anaesthetic identity as a thoroughly trained specialism, and one which was used to, and exerted a high level of situational control. When speaking about their training, participants commented positively. However, the desire for participants to extend and regularly repeat this thorough training in emergency procedures was equally recognised:

> *"I think regular putting yourself under that amount of pressure, even with the sim man, and regular sort of practice at performing the technical aspects of the procedure on the sim man but also in a person, when you're not under pressure, would probably help."* (LS34—PII)

The theme of 'Identity as an Anaesthetist' also encompassed the idea put forward by participants, that as a medical speciality, Anaesthetists are particularly used to high-level situational control, which was challenged by emergency, and therefore, unexpected situations. Multi-disciplinary working (such as that seen in this training programme) was stated as a possible solution to Anaesthetists being able to relinquish some situational control to better deal with emergency situations as they arise and also feel less challenged in their identity:

> *". . .it's been good doing the simulations with members of the theatre team because they're the people that you'll work with as well, and I think even though this is more aimed at training me it's helped them to know where things are. . ."* (LH32—RPII)

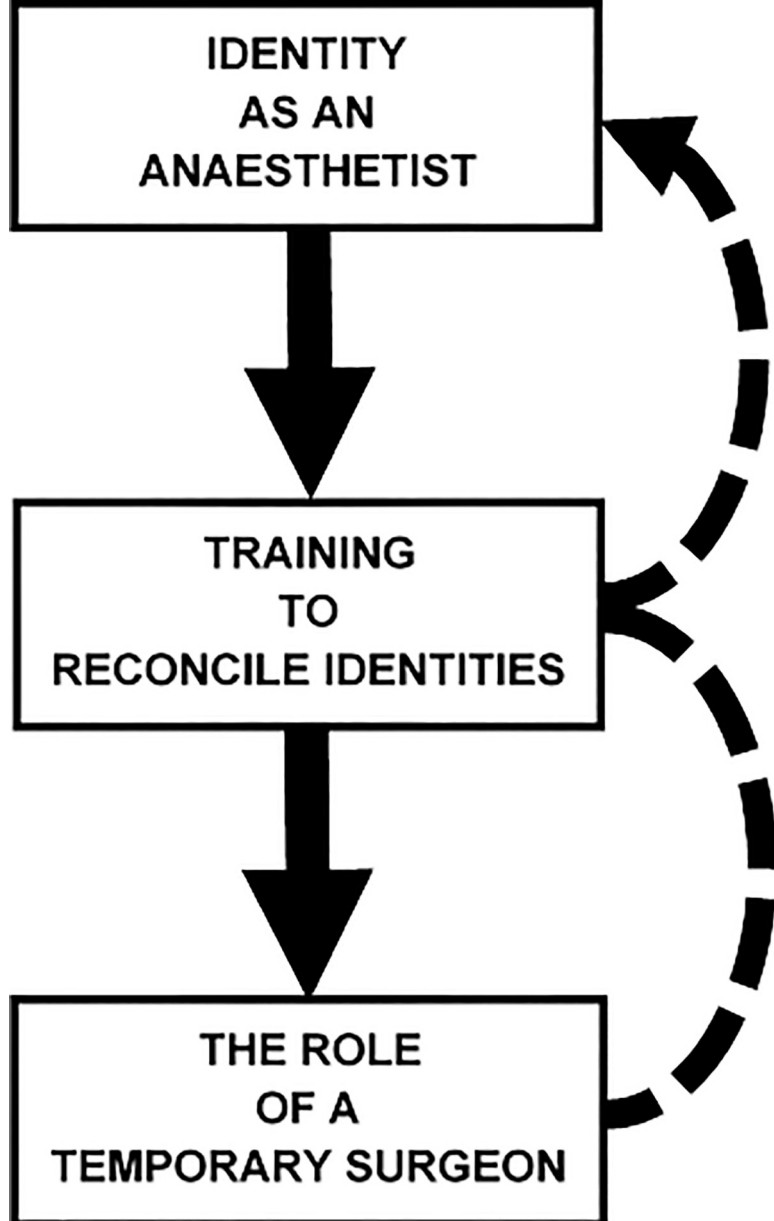

**Fig 3. Thematic diagram of themes.**

Likewise, many participants spoke of how the DAS Guidelines acted as helpful prompts to reinforce the training they had received, when dealing with difficult airways. However, some participants spoke of how unexpected surgical events, such as a CICO patient, affected them, but that these feelings of situational control as part of the Anaesthetic Identity prevailed even when patients elected to have airway procedures:

*". . .for an elective it's elegant, nice, controlled movements whereas for a Cric [sic] it's basically like. . . slash and stick a tube in."* (PE30—FPII)

Reflecting on their Identity as an Anaesthetic Trainee, participants stated how their training equipped them well for everyday procedures, but that they did not feel they could be confident

**Table 2. Supplementary quotations.**

| Theme 1: 'Identity as an Anaesthetist' | Theme 2: 'The Role of a Temporary Surgeon' | Theme 3: 'Training to Reconcile Identities' |
|---|---|---|
| *"I try and communicate with what's going on, just so people know what I'm thinking, 'cause sometimes I think people often tunnel into situations and often people don't want to say something–I think that's often a problem in emergencies. . ."* (LH32—PII) | *"I think my initial thoughts would be like, I can't believe this is happening to me, this isn't ever really supposed to happen. We're trained for it, but, you know, it's very, very rare that it happens. I'd be like, I can't believe this has happened to me. Where's the help?"* (LS34—PII) | *"So, accepting the fact that the patient that wasn't supposed to die now has a risk of dying because either you haven't done something, or you have done something wrong–you don't know at that point, so all that you need to sort of make that decision that actually 'yes, it is like this and I have to do something that I've never done before'."* (AS35—PII) |
| *"So, I think that's probably the hardest thing and having sometimes feeling quite overloaded, having lots of things to do, cognitively, you know, overloaded. Lots of things to do at once can be quite stressful."* (LS34—PII) | *". . .instinctively anaesthetists don't want to make big cuts in people because. . . sorry. . . because that's not what we do."* (SF36—PII) | *". . .the reluctance to finally pick up the scalpel, and my sort of thought before I go into this is the thought process of you have to have a healthy respect for going for it, but if you do decide to go for it you've got to go for it!"* (JH33—PII) |
| *"I was quite pleasantly surprised by the amount of training you get in anaesthetics as compared to when you're a foundation doctor. . ."* (CL31—PII) | *"I'm very well aware of the fact that human beings are fallible. And I know that, as much as I'd like to think that, I might not be able to perform as well as I would like to think, given the pressure of the situation that you're in, and that might affect patient outcome; and that frightens me."* (LS34—PII) | *". . .it's almost less urgent when his saturations are kind of normal to do something dramatic as to stick a big scalpel down his neck and stick tubes in his trachea, which obviously has a very high kind of complication risk. . . . . . . . . the key thing I suppose is declaring it to yourself that you can give yourself permission to go down those steps and then obviously communicate it to your team, 'cause you're gonna be the guy to hopefully get this patient out of a hole until somebody else comes to help."* (TW34—IPII) |
| *"I kind of see this as if you have to do it then you don't have a choice. So, we are very structured in the way we do our training and our anaesthetics, we have the DAS guidelines, we know what we are doing, and I think that sometimes there is an inability to accept that you have failed to get the airway."* (JH33—FPII) | *"I don't think we are programmed to accept failure, especially in something we do every day for how many years something you didn't plan for anyway and in a way, you are letting yourself down if you can't intubate that patient somehow."* (AS35—IPII) | *"I think it is the pressure that sort of stops you performing in real life, so it's good. I think it's a false way of creating a similar emotion to what you would feel if you were doing something like that in real life, so I like the fact that it made me feel a bit nervous, and I was able, I think largely because it was a manikin, work through that."* (LS34—IPII) |
| *"I think we've had a lot of repetitive practice and I think that is what has done that. It almost just flows now, like a robot."* (LS34—FPII) | *". . .that reality of saying that verbally to the team and almost admitting that you were out of options, and this is the only option left, I think is quite scary."* (TW34—IPII) | *"I think now I'd be more confident to do that with a bit more conviction, so I feel that even though that's an anxiety I've got a plan for that as well!"* (LH32—RPII) |
| | *"I think it's just everybody is desperate to avoid front of neck, and anyone who says there not is lying. . ."* (JH33—RPII) | *"I feel that I'm more better equipped to try and deal with that, and making sure that I take some time to position them to make that easier as well, but I suspect my two anxieties of them still getting the bougie and then the tube in would be perhaps more to a forefront than finding the membrane now."* (LH32—RPII) |
| | *"I think actually putting you know a sharp implement in there and making a cut on a human is a, is quite a hard thing to do. Particularly for us as anaesthetists because we don't really do it that much, apart from making a bigger track for a line, which is, you know."* (CL31—RPII) | *"The model is pretty realistic in that you've got blood spurting out. . . I think my fear was that in real life it might not be quite so straight forward. . . there was a lot of blood coming at me and I just managed to stay calm. . . I think that fear has kind of disappeared."* (CL31—FPII) |
| | *". . .obviously our oaths, the first thing is do no harm and I think it does feel a bit unnatural sticking a scalpel into someone's neck."* (MC34—FPII) | |
| | *"I will always have a fear about it. It is terrifying. You have to cut someone's neck, and if you don't they are going to die, so I am always going to have, it is always going to be a procedure that is going to bring about a massive stress response in whoever has to do it, no matter if it's a trainee or a consultant a year from retirement."* (JH33—FPII) | |

in performing rare procedures such as emergency FONA by SCT as they had not had enough training (prior to the training they received during this study):

> *"...we're very good at intubating patients and ventilating patients and if those things that we're very good and experienced at fail us then we have to move onto something we're not experienced in and I suppose we don't know whether we're good at or not because we haven't tried it so that's probably part of the fears associated with it..."* (SA32—FPII)

'Identity as an Anaesthetist' was spoken about extensively in the interviews. Participants' data concurs that the Anaesthetic specialism enables these Medical Professionals to be thoroughly trained, which leads to them being calm and proficient in their day-to-day undertakings; whilst being supported by a methodical and structured approach exemplified by the DAS Guidelines. The way in which participants spoke of their training also suggests it gives way to another facet of their identity–that of being able to experience a high level of situational control in their tasks when caring for a patient. From these interviews, it was evident that 'Identity as an Anaesthetist', was in fact challenged by emergency situations which, up until this training, they had not felt capable of carrying out, and in which they were no longer able to exert situational control. 'Identity as an Anaesthetist', is an important first step in this grounded theory, and forms the basis on which the other themes are built, and on which they–either directly, or indirectly–react.

### Theme 2: 'The Role of a Temporary Surgeon'

The second theme to be generated during the analysis of the interview data was that of 'The Role of a Temporary Surgeon'. This theme explains how participants viewed a CICO patient, and a subsequent emergency FONA by SCT procedure as forcing these participants to adopt 'The Role of a Temporary Surgeon', however reluctant they may have been to do so. Reluctance was eventually overcome by using preparatory techniques before anaesthetising patients as one participant explained in their second interview:

> *"I'd probably still say the kind of psychological side giving yourself permission to believe what your actions are doing, that you can't intubate, you can't kind of recognising that in your mind, and then giving yourself permission to go and slice open their neck, that's still quite a hurdle to get through!"* (TW34—IPII)

The theme of 'The Role of a Temporary Surgeon' was also explained through the analysis to incorporate participants' ideas that having to make this transition and perform an act of emergency surgery was a failure of their abilities in managing the airway effectively–which is normally something they do not have to face.

Adopting 'The Role of a Temporary Surgeon' positioned participants apart from their training as an Anaesthetist, becoming equally as responsible for this surgical procedure as their surgical colleagues when in theatre:

> *"...obviously, as an anaesthetist we are trained to perform emergency cricothyroidotomies, but if the patient actually belongs to the surgeon that is operating on that list and is an airways surgeon, essentially it's probably better to get them to do it because they're more used to doing these procedures..."* (MC34—RPII)

Moreover, it was not uncommon for participants to suggest that surgical procedures such as emergency FONA by SCT, were simply not a natural part of their duties. The idea of having

to abandon one's Identity as an Anaesthetist and become–even if only for a short period–a member of the surgical team, was described as almost a physical transition:

> ". . .you think your role is with the Anaesthetists, you know how you come in up to the point where they are about to start the trachea, becoming a part of another team. . . I suppose it wasn't as difficult to switch sides as I thought. Although once we had put the tracheostomy in, I then felt like I took my role back as an Anaesthetist, because I then did the bronchoscopy to check the position, which is what an Anaesthetist would do." (CL31—RPII)

The transition from Anaesthetist to Temporary Surgeon was also noted to have a psychological impact, inextricably linked to causing the patient harm when performing the life-saving procedure:

> "I think again it's going to be the psychological impact of switching in from something you didn't expect when you started to actually, now, I need to cut this patient's throat, to keep them alive, plus there's always, I think, the chance of the patient dying on you, which wasn't initially there, that creeps in slowly, so that increases anxiety levels and I think that's probably the biggest challenge to deal with anxiety." (AS35—FPII)

This concept of FONA as outside of the usual Anaesthetist role was often coupled with not wanting to cause harm to patients in their care. This became a dominant topic within this theme. In that respect, one participant rationalised how and why they would struggle to perform emergency FONA by SCT, should the responsibility be left to them in an emergency especially in relation to blood oxygen saturation (measured by pulse oximetry, where Anaesthetists aim for >96%, with any reading below 90% being a danger sign; commonly known by the acronym SATS) and use of Suxamethonium chloride (a muscle relaxant drug which Anaesthetists sometimes administer to patients to facilitate fitting a tube into their airway; commonly known by the acronym SUX):

> ". . .you struggle to grab a scalpel with SATS in the mid 90's, I think that is probably fair comment. Especially when there is another option, because the SUX can wear off, and that option ends with the patient just waking up and you apologising rather than that patient having their neck slashed open. . ." (JH33—FPII)

The theme of 'The Role of the Temporary Surgeon' demonstrates that whilst Anaesthetists know emergency FONA can be part of their duty to save a patient's life, the surgical role which they have to adopt is seen as paradoxically harmful to the patient and the decision to perform the emergency surgery is not easily made, even though they knew there was nothing else that could be done to the airway which would keep the patient alive:

> ". . .even though it's a life-saving procedure and you would do it if you had to, I think it's a bit more of an invasive procedure than perhaps we're used to doing as anaesthetists and you'd really want to know that this is like an end of the line thing before doing the procedure." (LH32—FPII)

Theme 2: 'The Role of a Temporary Surgeon' elaborates on participants' fears and concerns of causing harm to their patients whilst also highlighting the fact that reaching 'Plan D' of the DAS Guidelines is seen as a failure in managing the airway effectively, and therefore not fulfilling their role as an Anaesthetist who usually remain as the 'bloodless Doctor' in the operating

theatre. It is because of this, participants reportedly feel forced to make the, albeit temporary, transition and adopt a surgical role within the team, to intervene in the emergency situation and save the patient's life using emergency FONA by SCT. Prior to this training programme the barriers to effectively performing this emergency procedure were reported to be both psychological and physical with participants reporting uncertainty of both their competence and confidence in taking a surgical to a struggling patient's neck.

### Theme 3: 'Training to Reconcile Identities'

The third theme of this theory pertains to the training programme itself, in relation to the other two themes which are based on the identities of the participants transitioning between Anaesthetist and Temporary Surgeon in a time of crisis. Initially, the procedure was perhaps best summed up as:

> *"A life or death procedure, other options had been exhausted, so yeah unpleasant for everybody, but necessary in the end."* (JH33—PII)

Participants were, however, clear that the responsibility for this particular surgical procedure lay very much with the attending Anaesthetist. Enrolling in this training reinforced this point, which was beneficial in making sure these participants were both aware of the procedure and of their duty and their prerogative to perform it. Participants often reflected on their perceived level of competence when discussing if they might have to perform emergency FONA by SCT, but also reflected on gravity of the CICO situation with confidence and competence in the skill were spoken about at length. One participant spoke about bringing their level of competence to the fore of their conscious after the training:

> *". . .you're now conscious, you feel consciously incompetent, well hopefully more consciously competent, but certainly not expert. . . . . . . . It's a pretty major situation, and I think I'd be a bit arrogant to kind of suddenly assume I've got the surgical technique as well to do it competently in an emergency scenario anyway."* (TW34—IPII)

The training was also described by participants as being helpful to overcome some of the psychological barriers to performing it, and therefore heightening their perceived competence.

The training ultimately enabled participants to not only enhance their competence and confidence, but trained them to accept the need to perform this procedure, not as a failure of managing the airway, but as part of managing a failing airway successfully–and with this came the notion that they could give themselves permission to undertake such a procedure should the need for one occur:

> *". . .you can say to them I need to scan your neck because if I have trouble oxygenating you then you might wake up with a tube in your throat or a hole in your throat. And then you're almost giving yourself permission to do it before it's happened, psychologically."* (LS34—RPII)

On the whole, the training was seen as extremely beneficial for targeting the 'how to' part of the skill-base required for the procedure:

> *". . .thinking about what hand dominance you are which way to stand on which side of the patient and then cos you are kinda doing, you know ideally when you are slice through somebody's skin you want to be in control with your most dominant hand, but then your bougie, to*

*get the bougie in you don't really want to let go of your scalpel so working out what to do with your hands basically that was really useful thing."* (TW34—FPII)

With greater mastery of the technical skill, came greater confidence, which in turn aided a more proficient technical ability. Over the course of the six months of training, a major success noted by the team (see also Berwick et al., 2019) was that both confidence and competence grew, and there was a markedly more resounding acceptance of performing emergency FONA by SCT as a normal part of managing a failing airway The training was also seen to remove the anxieties surrounding 'Plan D' of the DAS Guidelines, and normalise it as an anaesthetic procedure:

*"I'd feel less anxious about actually having to do it for the person in real life and less 'Oh God is this actually happening to me?' and just be like, 'Well, I've done what I can so far, I can't intubate them, definitely can't ventilate them and the next step is this. . .'. Hopefully it would be less of a barrier to actually perform that next step after having done all the practice and all the teachings and stuff now. So, I think if I had to do it in real life, I would probably be more confident about it and less hesitant."* (LS34—FPII)

The training was reported by participants to not only elevate self-reported confidence, and competence in the skill of, and actually performing emergency FONA by SCT, but also reinforcing to participants that the procedure is necessary and permissible, with it in fact being recommended in the DAS Guidelines, thus allowing for acceptance of the procedure as part of an Anaesthetists arsenal of techniques to effectively manage an airway. The training also allowed for participants to reconcile their Identity as an Anaesthetist with the need for them to become a Surgeon temporarily for the period of time it took to carry out the procedure before transitioning back into the Anaesthetist role once again. From the data, it can be see this comprehensive training programme enabled participants to feel able to smoothly transition into taking on 'The Role of a Temporary Surgeon' with both the confidence and the technical competency to carry out emergency FONA by SCT, but also, and perhaps more importantly, this 'Training to Reconcile Identities' offered participants acceptance of the procedure which allowed them to return to their 'Identity as an Anaesthetist' without the psychological distress about the procedure and situation itself (i.e. causing harm to the patient and failing to maintain an airway), they had all once reported prior to the training.

## Discussion

Emergency FONA represents a high-pressure, life-threatening scenario, even to experienced Anaesthetists. Whilst fortunately rare, outcomes following a CICO can be disastrous, and therefore all Anaesthetists, not just Specialist Trainees, are required to be trained in its management. To date, this has largely taken the form of simulation and the rehearsal of technical skills on manikins and cadaveric pro-sections. Historically, anaesthetic training in the UK has focussed on developing technical skills and clinical decision making. More recently there has been increased attention on the importance of human factors, non-technical skills, in clinical practice, especially in emergency situations and multi-disciplinary training [6, 8, 37].

Grounded Theory, to date, has not been widely used in the assessment or development of training in Anaesthesia, nor in clinical education more widely. This study demonstrates the utility of Grounded Theory to identify psychological elements surrounding the management of a CICO scenario. The generation of our three-part theoretical model (*see* Fig 3) provides two ingredients for the development of a more comprehensive and efficacious approach to

training for CICO and emergency FONA by SCT, in particular. Firstly, it gives us a greater understanding of the challenges faced by Anaesthetists confronted by this situation and, in doing so, we can better tailor future training to fit the needs of Anaesthetists. Secondly, with this better developed awareness of training needs we can more effectively evaluate the training programmes currently in use against ours and map the provisions of our training regime against the recognised technical and non-technical needs of those being trained.

Looking more closely at our themes of 'Identity as an Anaesthetist' and 'The Role of a Temporary Surgeon' we can better understand the decision-making processes taken during a CICO scenario. The majority of participating Anaesthetists in this study described the identity of the 'Anaesthetist' as that of a highly-trained individual, used to a high degree of situational control, and with the ability to remain calm and methodical, especially when confronted by in emergency situations [38–41]. Anaesthetists usually expect other team members to 'look to them' for leadership and tend to adopt a 'command-and-control' style [6, 37, 42–46]. Reaching a CICO scenario was seen almost universally as a failure to provide the usual standard of care to patients, despite being clearly stated as 'Plan D' in the DAS guidelines [1]. Indeed, throughout the interviews participants rarely described FONA as 'life-saving' with the language used to describe FONA being universally negative. There was also a high level of discomfort at the perceived loss of situational control, and a degree of disbelief regarding the evolving situation. The requirement to use a scalpel and perform an 8-10cm incision (for impalpable cricothyroid membranes) was a known, but unfamiliar procedure to all participants and seen as far removed from the usual procedures Anaesthetists carry out as part of their routine work. Therefore, even as a 'last resort' to save a patient's life, trainees remained averse to it. This transition to 'temporary Surgeon' was linked to the perception of participants failure as an Anaesthetist, who usually remain as the 'bloodless Doctors' in the operating theatre, and who are now suddenly seeing themselves as having to cause harm to the patient, despite it being a lifesaving procedure. An Anaesthetist must transition between the roles of 'Anaesthetist' and 'Surgeon' to successfully accomplish FONA, but the training provided enabled them to transition back into the role of an 'Anaesthetist' once the airway had been secured (i.e. once an endotracheal tube has been successfully positioned) and the patient was again stabilised. The illumination of how participants viewed these roles makes it clear this transition is not a straightforward one and more than just the wielding of a surgical instrument. It is the movement from a role of control and responsibility, through self-described 'failure' and into a role of uncertainty about one's own skill competence and confidence.

To tailor a training programme to support this transition from Anaesthetist to 'temporary Surgeon', and back to that of Anaesthetist requires the mapping of training to the challenges of the procedure. We need to address technical skills for improving confidence in the new surgical role, but also incorporate concepts such as resilience into training [40]. By acknowledging feelings of anxiety and so-called failure as 'normal', and training to 'live with' the feeling whilst continuing to manage the CICO scenario in a calm and controlled manner, we may improve confidence and decrease time to decision made (for early appraisals of this possibility see Berwick and colleagues [6]; and Groom and colleagues [8]). In using Grounded Theory to evaluate our current training model we find much that is helpful whilst recognising areas which can be strengthened. The interviews demonstrate training served to increase candidate's confidence and competence in performing emergency FONA using SCT, smoothing the transition from Anaesthetist to Surgeon and, most importantly, back again, thus reconciling these two, seemingly opposing identities. It also altered candidate's perceptions of emergency FONA by SCT from that of so-called failure as an Anaesthetist to a necessary and natural progression in successful and correct management of a difficult airway. This was maintained at 6-months following the training.

There are still areas which may warrant revisiting in light of the Grounded Theory produced from this study. Training is still overtly focused on the mastery and practice of technical skills which was addressed by the *in-situ* multi-disciplinary team simulations which provided opportunities to rehearse and refine technical skills of the CICO emergency. It may, however, be that more direct attention is required to address the anxiety and discomfort around this emergency procedure. Whilst no training tool will ever obfuscate worry and stress at a life-threatening situation, we can, and must, do better to train for the real-world issues now we have a better understanding of them.

## Strengths and limitations

This study has many strengths. Firstly, we have demonstrated that this joint anaesthetic and surgical training programme, which was the first of its kind specifically for emergency FONA by SCT, can be successfully implemented and evaluated. Secondly, and to the authors' knowledge, this is the first time Grounded Theory has been used in the context of professional identity for Anaesthetists. Further strengths can be found in how the findings presented in this article have benefitted from a rigorous approach to the design, data collection, and analysis. Using a data-driven approach to analysis has ensured our data remained true to the experiences of our participants and the inter-disciplinary nature of the research team has enabled us to thoroughly interrogate our data to ensure it satisfies the competing demands of clinical practice, medical education, and psychological theorisation on professional identity.

A limitation of this study is the Specialist Trainee Anaesthetists we recruited were all from one UK NHS Hospital, and therefore may not be generalisable to Specialist Trainee Anaesthetists in other UK Trusts, or indeed those working in different countries. This could also apply to the medico-legal contexts of different countries, where fear of litigation may be more commonplace than in others. We believe this limitation is mitigated by the fact that this analysis was based on the evaluation of a pilot of the training programme we devised, and so was small by design. Yet, our rigorous longitudinal data collection approach (during which we maintained a 100% retention rate), has meant we could achieve rich and complete datasets, as well as theoretical saturation, with few participants.

In summary, we suggest our study has many more strengths than limitations and whilst contributing new theory to the lacuna in Anaesthetic training has been an important endeavour, we advocate most strongly the use of Grounded Theory in cross-disciplinary health research, medical education, and clinical training.

## Implications for clinical practice and future research

Given the impossibility of designing a study which involves a transfer of training to a real-life CICO situation, we believe this training programme has been an effective way of preparing Anaesthetists for this rare event. Whilst we would not expect every hospital to be able to deliver this training, scale-up is possible, as long as the key aspects of this training (such as the inter-disciplinary arrangements) can be met. Alternatively, specialist centres could offer this training for all trainees within a particular region, in-line with other specialist training currently available to medical professionals. We have also demonstrated that an inter-disciplinary training and research programme can be successfully implemented, and suggests how to overcome the issues of inter-disciplinary working which are frequently cited in the literature [12, 47, 48]. We have further demonstrated the utility and value of Grounded Theory in clinical and medical education and cross-disciplinary health research. We therefore suggest that, as with all Grounded Theory, this analysis has produced a testable hypothesis, and would implore research teams in different settings, to test our theory with their populations. We would also

recommend researchers undertake a Grounded Theory analysis of Anaesthetists' retrospective accounts of dealing with a real-life CICO situation, which would add greater perspective on this incredibly rare, but life-threatening clinical event.

## Conclusions

The importance of training in medical practice cannot be overstated. It forms the foundations of safer and more efficient practice as well as the development of excellence. For training to be effective it must be able to deliver what the trainee needs in order to improve their skills, overcome obstacles or advance their knowledge of a task. CICO situations do not provide the luxury to plan, therefore training Anaesthetists to adopt a surgical role when required, provides them with not only the skills and competence, but the confidence to assess who is suitably qualified to perform the emergency FONA procedure. We now have the means to better understand what challenges stand between the trainee and a task and in doing so can more effectively match training to task. We conclude by stating our belief that Grounded Theory has a much wider role to play in future research to understand the complexity of interventions in medical education.

## Acknowledgments

The authors would like to extend their sincere thanks to all the participants who gave their time to take part in the original study during their training, without whom this research would not have been possible. We would also like to extend our gratitude to Dr. Melissa Pilkington (Manchester Metropolitan University) for her occasional assistance in collecting interview data.

## Author Contributions

**Conceptualization:** Sergio A. Silverio, Richard Berwick, Simon Mercer, Simon N. Rogers, John E. Sandars, Peter Groom, Jeremy M. Brown.

**Data curation:** Sergio A. Silverio, Jeremy M. Brown.

**Formal analysis:** Sergio A. Silverio, Hilary Wallace, William Gauntlett, Jeremy M. Brown.

**Funding acquisition:** Peter Groom.

**Investigation:** Sergio A. Silverio, Hilary Wallace, William Gauntlett, Jeremy M. Brown.

**Methodology:** Sergio A. Silverio, Hilary Wallace, William Gauntlett, Jeremy M. Brown.

**Project administration:** Sergio A. Silverio, Hilary Wallace, William Gauntlett, Richard Berwick, Peter Groom, Jeremy M. Brown.

**Resources:** Sergio A. Silverio, Richard Berwick, Simon Mercer, Simon N. Rogers, Peter Groom.

**Software:** Sergio A. Silverio.

**Supervision:** Sergio A. Silverio, Simon Mercer, Simon N. Rogers, Peter Groom, Jeremy M. Brown.

**Validation:** Sergio A. Silverio, Peter Groom, Jeremy M. Brown.

**Visualization:** Sergio A. Silverio.

**Writing – original draft:** Sergio A. Silverio, Hilary Wallace, William Gauntlett, Jeremy M. Brown.

**Writing – review & editing:** Sergio A. Silverio, Hilary Wallace, William Gauntlett, Richard Berwick, Simon Mercer, Ben Morton, Simon N. Rogers, John E. Sandars, Peter Groom, Jeremy M. Brown.

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
