## [Decision Letter · Decision Letter 0]

15 Dec 2020

PONE-D-20-24393

Becoming the temporary surgeon: A grounded theory examination of anaesthetists performing emergency front of neck access in inter-disciplinary simulation-based training.

PLOS ONE

Dear Dr. Silverio,

Thank you for submitting your manuscript to PLOS ONE. After careful consideration, we feel that it has merit but does not fully meet PLOS ONE’s publication criteria as it currently stands. Therefore, we invite you to submit a revised version of the manuscript that addresses the points raised during the review process.

Please address the reviewers' comments The manuscript is too lengthy. It would be easier to be followed if it has been abstracted. 

This decision is based on PLOS ONE’s publication criteria.

We look forward to receiving your revised manuscript.

Kind regards,

Mohamed R. El-Tahan, MD

Academic Editor

PLOS ONE

Journal Requirements:

2. It is unclear why 'No - some restrictions will apply' ; please clarify the nature of these restrictions, ie. If due to ethical or legal reasons.

PLOS only allows data to be available upon request if there are legal or ethical restrictions on sharing data publicly. For information on unacceptable data access restrictions, please see http://journals.plos.org/plosone/s/data-availability#loc-unacceptable-data-access-restrictions.

3. Thank you for your ethics statement:

'Ethical approvals were sought from and granted by Health Research Authority with IRAS project ID: 226720 and REC reference: 18/HRA/0122, with both University and Hospital permissions, subsequently granted.'

4. Please provide additional details regarding participant consent.

In the ethics statement in the Methods and online submission information, please ensure that you have specified (i) whether consent was informed and (ii) what type you obtained (for instance, written or verbal, and if verbal, how it was documented and witnessed). If your study included minors, state whether you obtained consent from parents or guardians. If the need for consent was waived by the ethics committee, please include this information.

'Sergio A. Silverio (King’s College London) is supported by the National Institute for Health Research Applied Research Collaboration South London [NIHR ARC South London] at King’s College Hospital NHS Foundation Trust.  The views expressed are those of the authors and not necessarily those of the NIHR or the Department of Health and Social Care.  '

a. Please confirm that this does not alter your adherence to all PLOS ONE policies on sharing data and materials, by including the following statement: "This does not alter our adherence to  PLOS ONE policies on sharing data and materials.” (as detailed online in our guide for authors http://journals.plos.org/plosone/s/competing-interests).  If there are restrictions on sharing of data and/or materials, please state these.

Please note that we cannot proceed with consideration of your article until this information has been declared.

6. Please upload a copy of Figure 3, to which you refer in your text on page 16. If the figure is no longer to be included as part of the submission please remove all reference to it within the text.

Additional Editor Comments:

Dear Authors

Thank you very much for your interest in the Plos One. The two expert reviewers have raised interesting suggestions to improve the quality of the manuscript. Can you please kindly address all comments in the next version of the manuscript. I would highlight the followings

1. The generalizability and logistics for including FONA in the curriculum of training in anaesthesia.

2. The need to reduce the length of the manuscript.

Reviewers' comments:

Reviewer's Responses to Questions

**Comments to the Author**

1. Is the manuscript technically sound, and do the data support the conclusions?

Reviewer #1: Yes

Reviewer #2: Yes

2. Has the statistical analysis been performed appropriately and rigorously? 

Reviewer #1: N/A

Reviewer #2: Yes

3. Have the authors made all data underlying the findings in their manuscript fully available?

Reviewer #1: Yes

Reviewer #2: Yes

4. Is the manuscript presented in an intelligible fashion and written in standard English?

Reviewer #1: Yes

Reviewer #2: Yes

5. Review Comments to the Author

Reviewer #1: Thanks for this interesting study and study approach;

however, there are some concerns.

I do support the idea to use structured interviews and identify frames of action the study participants. However the manuscript has an incredible length; as well documented, adults (and even kids with different devices) can use scales to grade certain expressions describing feelings / ideas etc. this might help to grade the progress in the training. The bites from the interviews are interesting to read, but too long ; you should put those in a supplemental file

I understand that this longitudinal study in this setting cannot be designed to train hundreds of residents. Is you approach applicable for larger groups?

Please straighten the manuscript; it is way too long; this accounts for all parts of the document.

Reviewer #2: Congratulations for the very interesting study.

I have just one suggestion, although it is very probably not even fairly related to the context.

First of all, the addition of FONA into the curriculum of anaesthesia should be set a precedent in other countries. During the process I think one of the most important steps is medico-legal and more importantly "self-ethical" issues. (To explain my concern: If an anaesthesist has to apply a FONA, she/he should have no concerns about the possible question "if I do something wrong, will I be sued?"; and in the second , more important step, the more important question: "if I do something wrong, will I ask myself that it would be better to let the surgeon do it?"....

I think these questions deserve also to be discussed.

6. PLOS authors have the option to publish the peer review history of their article (what does this mean?). If published, this will include your full peer review and any attached files.

Reviewer #1: No

Reviewer #2: No

---

## [Author Response · Author response to Decision Letter 0]

29 Jan 2021

Wednesday 27th January 2021

Dear Prof. El-Tahan, Academic Editor of PLoS One,

Re:- Re-Submission of Research Article [PONE-D-20-24393] with Revisions.

We attach for your consideration a revised version of our article titled: ‘Becoming the temporary surgeon: A grounded theory examination of anaesthetists performing emergency front of neck access in inter-disciplinary simulation-based training.’

We would like to extend our sincere thanks to you and the Reviewers for such positive commentaries on our manuscript and for the helpful proposed revisions. We have carefully considered these revisions and have made changes accordingly and appropriately.

You will find attached the revised manuscript with revisions inserted in track changes, as well as a separate document containing a table with Editors’ and Reviewers’ comments and our corresponding responses stating how and where we have implemented the suggested changes.

We are extremely grateful for the opportunity you have provided us to revise and re-submit this paper, and believe your reviews and our changes have made the paper stronger.

Yours Faithfully,

Sergio A. Silverio 

MPsycholSci (Hons) L'pool, MSc Brun, AFBPsS, FRSPH, FRAI, FMHC, RSci

Research Associate in Social Science of Women’s Health, Department of Women & Children’s Health, King’s College London

and on behalf of 

Hilary Wallace - Anaesthesia and Theatres Department, Liverpool University Hospitals NHS Foundation Trust

William Gauntlett – The Jackson Rees Department of Anaesthesia, Alder Hey Children’s NHS Foundation Trust

Richard Berwick - Pain Research Institute, University of Liverpool

Simon Mercer - Medical Education Department, Liverpool University Hospitals NHS Foundation Trust

Ben Morton - Department of Clinical Sciences, Liverpool School of Tropical Medicine

Simon N. Rogers - Oral and Maxillofacial Surgery Department, Liverpool University Hospitals NHS Foundation Trust

John E. Sandars - Health Research Institute, Edge Hill University

Peter Groom - Anaesthesia and Theatres Department, Liverpool University Hospitals NHS Foundation Trust

Jeremy M. Brown - Health Research Institute, Edge Hill University

---

## [Decision Letter · Decision Letter 1]

11 Mar 2021

Becoming the temporary surgeon: A grounded theory examination of anaesthetists performing emergency front of neck access in inter-disciplinary simulation-based training.

PONE-D-20-24393R1

Dear Dr. Silverio,

We’re pleased to inform you that your manuscript has been judged scientifically suitable for publication and will be formally accepted for publication once it meets all outstanding technical requirements.

Kind regards,

Mohamed R. El-Tahan, MD

Academic Editor

PLOS ONE

Additional Editor Comments (optional):

Reviewers' comments:

Reviewer's Responses to Questions

**Comments to the Author**

1. If the authors have adequately addressed your comments raised in a previous round of review and you feel that this manuscript is now acceptable for publication, you may indicate that here to bypass the “Comments to the Author” section, enter your conflict of interest statement in the “Confidential to Editor” section, and submit your "Accept" recommendation.

Reviewer #1: All comments have been addressed

2. Is the manuscript technically sound, and do the data support the conclusions?

Reviewer #1: Yes

3. Has the statistical analysis been performed appropriately and rigorously? 

Reviewer #1: Yes

4. Have the authors made all data underlying the findings in their manuscript fully available?

Reviewer #1: Yes

5. Is the manuscript presented in an intelligible fashion and written in standard English?

Reviewer #1: Yes

6. Review Comments to the Author

Reviewer #1: Dear authors, thanks for the work in the mausncript. I really appreciate the table with the statements- it gives now a good overview.

Kindes regards Christian Brülls

7. PLOS authors have the option to publish the peer review history of their article (what does this mean?). If published, this will include your full peer review and any attached files.

Reviewer #1: No

---

## [Editor Report · Acceptance letter]

15 Mar 2021

PONE-D-20-24393R1 

Becoming the temporary surgeon: A grounded theory examination of anaesthetists performing emergency front of neck access in inter-disciplinary simulation-based training. 

Dear Dr. Silverio:

I'm pleased to inform you that your manuscript has been deemed suitable for publication in PLOS ONE. Congratulations! Your manuscript is now with our production department. 

Kind regards, 

on behalf of

Professor Mohamed R. El-Tahan 

Academic Editor

PLOS ONE